# Optimization of Electrical Intensity for Electrochemical Anodic Oxidation to Modify the Surface of Carbon Fibers and Preparation of Carbon Nanotubes/Carbon Fiber Multi-Scale Reinforcements

Mengfan Li [1,2], Yanxiang Wang [1,2,*], Bowen Cui [1,2], Chengjuan Wang [1,2], Hongxue Tan [1,2], Haotian Jiang [1,2], Zhenhao Xu [1,2], Chengguo Wang [1,2] and Guangshan Zhuang [1,2,*]

1   Key Laboratory of Liquid-Solid Structural Evolution and Processing of Materials of Ministry of Education, Shandong University, Jinan 250061, China
2   Carbon Fiber Engineering Research Center, School of Material Science and Engineering, Shandong University, Jinan 250061, China
*   Correspondence: wangyanxiang079@163.com (Y.W.); zhuangguangshan01@163.com (G.Z.)

**Abstract:** Carbon fiber (CF) reinforced composites are widely used due to their excellent properties. However, the smooth surface and few functional groups of CFs can lead to fiber fractures and pullout, which reduce the service life of the composites. The overall performance of composites can be improved by growing carbon nanotubes (CNTs) on the CF surface. Before this, CF surface should be modified to enhance the loading amount of catalyst particles and thus make the CNTs more uniform. In this paper, CNTs were grown on a CF surface by one-step chemical vapor deposition to prepare multi-scale CNTs/CF reinforcements, and the effects of different methods on the CF surface modification were explored. After setting four intensities of electrochemical anodic oxidation, i.e., 50 C/g, 100 C/g, 150 C/g and 200 C/g, it was found that the distribution and quantity of CNTs were improved under both the 100 C/g and 150 C/g conditions. Considering the influence of electrical intensity on the (002) interplanar spacing of CFs, which affects the mechanical properties of the samples, 100 C/g was finally selected as the optimal electrochemical treatment intensity. This finding provides a reference for continuous and large-scale modification of CF surfaces to prepare CNTs/CF multi-scale reinforcements.

**Keywords:** carbon nanotube; carbon fiber; surface modification; electrochemical anodic oxidation

## 1. Introduction

Carbon fiber (CF) has excellent properties, including high tensile strength, high tensile modulus, good compressive strength, low density, and excellent electrical and thermal/chemical resistance [1]. CF reinforced polymer composites have been widely used in aerospace, military industry, transportation, leisure and sports, and other fields [2–4]. These composite materials are made of resin and CFs by laminating and pressing under high temperature and pressure. Resin, as a continuous phase, and CFs, as a discontinuous phase, both contribute to the overall performance of the composite materials [5]. However, the contribution of the resin matrix to the overall performance of the composite materials is limited and cannot be easily improved. Therefore, the improvement of the strength of composite materials mainly depends on the performance of the interface between the resin and fibers, and enhancements of interface performance are usually achieved by the surface modification of CFs [6–8]. Common CF surface modification treatment methods include the chemical grafting method [9,10], chemical vapor deposition (CVD) method [11–13], etc. Among them, the method of in situ growth of CNTs on the surface of CFs by CVD has received a great deal of attention. This method has the advantages of adjustable process parameters, controllable growth of CNTs, economical-growth equipment, etc. [14–17]. Moreover, as a

nanomaterial, CNTs have a special one-dimensional tubular structure, and their mechanical properties, electrical and thermal conductivity, are marvelous [18,19]. Therefore, growing CNTs on the CF surface to prepare CNTs/CF multi-scale reinforcements can introduce the excellent properties of CNTs into CF reinforced composites, resulting in a good reinforcement effect [20–23]. Ideal CNTs/CF multi-scale reinforcements need to uniformly grow a layer of CNTs on the surface of CFs. The question of whether catalyst particles, as the "seeds" for growing CNTs, can achieve firm attachment and uniform distribution is crucial in obtaining high-performance samples [24]. Nevertheless, as an industrialized product, CF has undergone pre-oxidation and medium-high temperature carbonization treatment. The surface is composed of an ordered, six-membered ring graphite sheet structure with almost no active functional groups. The surface is smooth and chemically stable [25,26], which prevents the attachment of a catalyst precursor and gives rise to the phenomenon of uneven distribution or agglomeration and deactivation of catalyst particles. Hence, the surface of CFs must be modified to make it easier to attach a catalyst precursor in order to ensure the uniform growth of CNTs. Common surface treatment methods for CFs include electrochemical anodic oxidation (EAO) [27–29], plasma treatment [30,31], liquid oxidation [32], etc. Among these, EAO and the liquid oxidation methods have simple operation processes and low economic cost and are worthy of in-depth research and development [27–29,32]. In particular, since CF has good chemical stability in various aqueous electrolytes and excellent electric conductivity, EAO can be used as an ideal surface modification approach to treat continuous fibers [33]. Additionally, many parameters can be adjusted during EAO, such as current, potential, and electrolyte composition [33,34]. Moreover, the treatment process of EAO is relatively mild and will not cause significant damage to the fiber surface [35]. Generally speaking, EAO includes two processes: anodic oxidation and cathodic reduction. The input electric energy is converted into chemical energy, and water is decomposed into strong oxidants at the anode to etch the surface of CFs, causing changes in surface functional groups and roughness [36,37].

In this study, the effects of EAO and liquid oxidation on the surface modification of CFs were compared. The EAO process was determined to be the most suitable method for the production of CNTs/CF multi-scale reinforcements. The process was then optimized by adjusting the electrical intensity.

## 2. Materials and Methods

### 2.1. Materials

PAN-based CFs (T-700-12k) were provided by Toray Co. Ltd., Tokyo, Japan. Nitric acid ($HNO_3$, 68%), hydrogen peroxide ($H_2O_2$, 30%), ethanol ($C_2H_5OH$, 99.7%), and ammonium dihydrogen phosphate ($NH_4H_2PO_4$, 99%) were purchased from Sinopharm Co. Ltd., Shanghai, China. Nickel nitrate ($Ni(NO_3)_2 \cdot 6H_2O$, 99.9%) was supplied by Aladdin Reagent Co. Ltd., Shanghai, China. $N_2$, $H_2$ and $C_2H_2$ were purchased from Jinan Gas Factory, Jinan, China. Deionized water was self-made in the laboratory.

### 2.2. Preparation of CNTs/CF Multi-Scale Reinforcements

The surface of the finished CFs adheres to the sizing agent. The CFs need to be placed in a vertical CVD furnace in advance and kept at 450 °C for 1.5 h in $N_2$ atmosphere. After removing the sizing agent on the surface of CFs, the desized CFs are obtained.

The surface modification treatments of desized CFs were carried out by $HNO_3$ treatment, $H_2O_2$ oxidation, and EAO, respectively.

- As for $HNO_3$ treatment, the desized CFs were fully immersed in a solution tank filled with $HNO_3$ aqueous solution (10 wt%), which kept in a 60 °C oven for 1 h.
- With regard to $H_2O_2$ oxidation, we fully submerged the desized CFs into a solution tank filled with $H_2O_2$ aqueous solution (30 wt%) and kept them in a 60 °C oven for 1.5 h.
- Regarding EAO, this study adopted a laboratory-made EAO treatment device, as shown in Figure 1a. The desized CFs entered the electrolytic tank through the guide

roller, and the electrolyte was an aqueous solution of $NH_4H_2PO_4$ (5 wt%). The two variables affecting the etching degree of the CF surface, i.e., the wire speed and the electrolytic intensity, were integrated into the electrical intensity per unit mass of CFs in order to quantify the intensity of the EAO treatment.

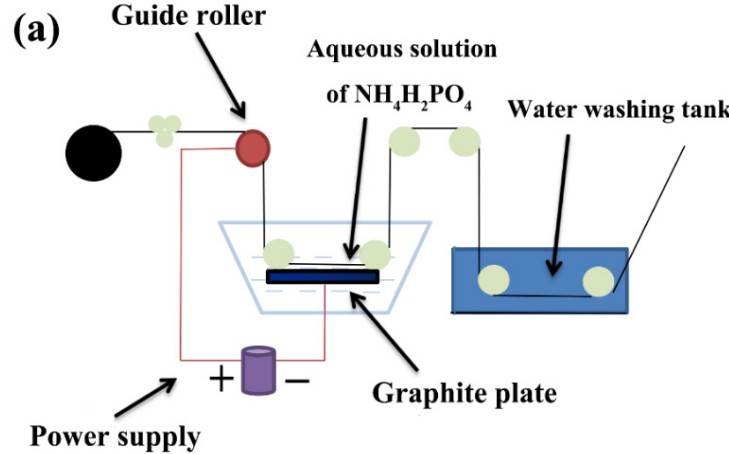

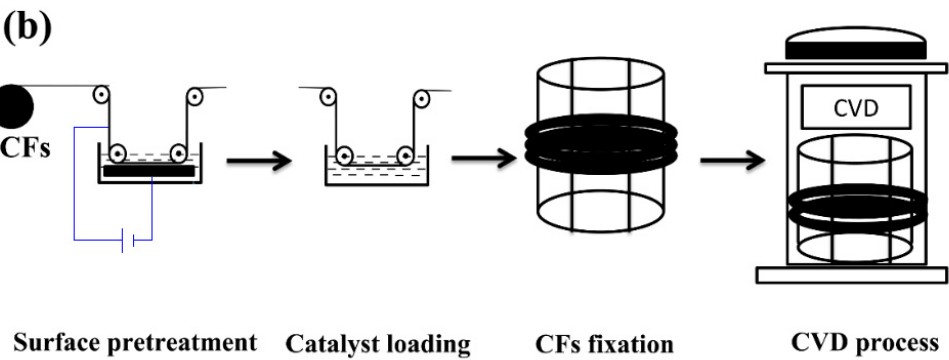

**Figure 1.** (**a**) EAO device, (**b**) Schematic diagram of CNTs grown on the CF surface.

Using the solution impregnation method, the catalyst solution was a 0.05 M $Ni(NO_3)_2$ alcohol solution. We then put the CFs in a drying oven at 70 °C. A one-step method was used to grow CNTs on the CF surface. The dried CFs were then placed into a vertical CVD furnace, which was heated to a reduction temperature of 450 °C, and $H_2$ was introduced to reduce the catalyst precursor to elemental Ni, before being adjusted to 550 °C. Additionally, 0.3 L/min $H_2$, 0.6 L/min $N_2$, and 0.3 L/min $C_2H_2$ were injected and kept for a long enough time to make CNTs grow on the surface of CFs. The CVD process in this device was kept in a quasi-vacuum state and a $N_2$ atmosphere. Figure 1b is a schematic diagram of the process, from CF surface modification treatment to the final growth of the CNTs.

*2.3. Characterization*

The morphological characteristics of CNTs/CF multi-scale reinforcement samples under different conditions were observed by scanning electron microscope (SEM, SU-70). After the surface modification treatment, the chemical composition and content of CF surface were analyzed by X-ray photoelectron spectroscopy (XPS, Thermo Fisher ESCALAB 250). Changes in the crystal structure of the samples were tested by X-ray diffraction (XRD, Rigaku D/max-RC).

The single-filament tensile strength of the prepared samples was tested according to ASTM D3822-07 standard. At least 40 CF filaments were prepared for each sample, and the average value was taken as the single-filament tensile strength of a single sample after testing in sequence.

## 3. Results and Discussion

### 3.1. Effect of Surface Modification on CF Surface Morphology

Figure 2 shows SEM images of the samples with different surface modification treatments, the surfaces after growing CNTs, and a structure diagram of CNTs/CF multi-scale reinforcements. It can be seen from Figure 2a,b that the surfaces of the desized CFs were very smooth and that CNTs had grown on the surface after the CVD process. However, the distribution was extremely uneven. Many CFs were not covered with CNTs to expose the smooth surface, and a large number of impurities and clustered CNTs appeared in the gaps between adjacent CFs. This was because the unmodified CF surface could not attract the catalyst precursor particles in solution effectively, meaning that it was left in the gaps. In this case, the catalyst precursor remaining after the volatilization of anhydrous ethanol was prone to agglomeration to form coarse catalyst particles during reduction. Additionally, local aggregations of CNTs formed during the catalysis process, and the catalyst particles that were too large lost their activity. In contrast, in Figure 2c–h, it can be seen that the surface of samples underwent slight changes after the modification treatment. This was due to the surface of CFs being oxidized and etched to make an unstable and disordered turbostratic graphite structure which fell off, leaving an indentation [38]. In comparison, the grooves on the surface of CFs treated with $HNO_3$ solution were too thin and shallow. Additionally, the inconspicuous grooves made a limited contribution to the adhesion of the catalyst particles, the distribution was not uniform, and there were point defects on the surface. The surface of EAO-treated samples had a relatively obvious groove structure, which was evenly distributed, and the groove morphology was better than those of the other two methods. The obvious groove morphology showed that its width and depth were large. Such a structure can effectively provide a "landing field" for precursor particles. The effect of $H_2O_2$ treatment was somewhere in between. The morphology images of the grown CNTs also effectively reflect the role of the grooves. Compared with the desized CFs, the growth quantity of CNTs in the $H_2O_2$-treated sample was not significantly increased, and the distribution uniformity was slightly improved. The number of CNTs in the $HNO_3$-treated sample was significantly increased, but the CNTs only grew on half of the sides of CF, presumably because the CFs were not under tension during the immersion modification treatment, and some CF surfaces were not fully etched by lamination. The effect of CNTs/CF obtained by EAO treatment was most satisfactory; the CNTs were evenly distributed and the density was appropriate.

The growth mechanism of CNTs on the CF surface is shown in Figure 2i. Firstly, the catalyst particles attached to the modified CF surface, and then the activated carbon atoms formed by the cracking of the carbon source gas diffused through the interior or surface of the catalyst particles. Finally, after being closely connected with the CF surface through chemical bonds, the remaining activated carbon atoms gathered together to grow hollow CNTs in a "top growth" mode. CFs, CNTs connected with them, and catalyst particles together constitute CNTs/CF multi-scale reinforcements.

### 3.2. Effect of Surface Modification on Single-Filament Tensile Strength

During the preparation of CNTs/CF multi-scale reinforcements, CFs will be damaged to different degrees, which will affect the mechanical properties of the material. The CF surface modification pretreatment process essentially destroys the surface structure of CFs and etches the trench structure. As such, the decrease in strength should be minimized while achieving a certain effect. The modified CFs obtained by different treatment methods were used for a single-filament tensile test; the results are shown in Figure 3.

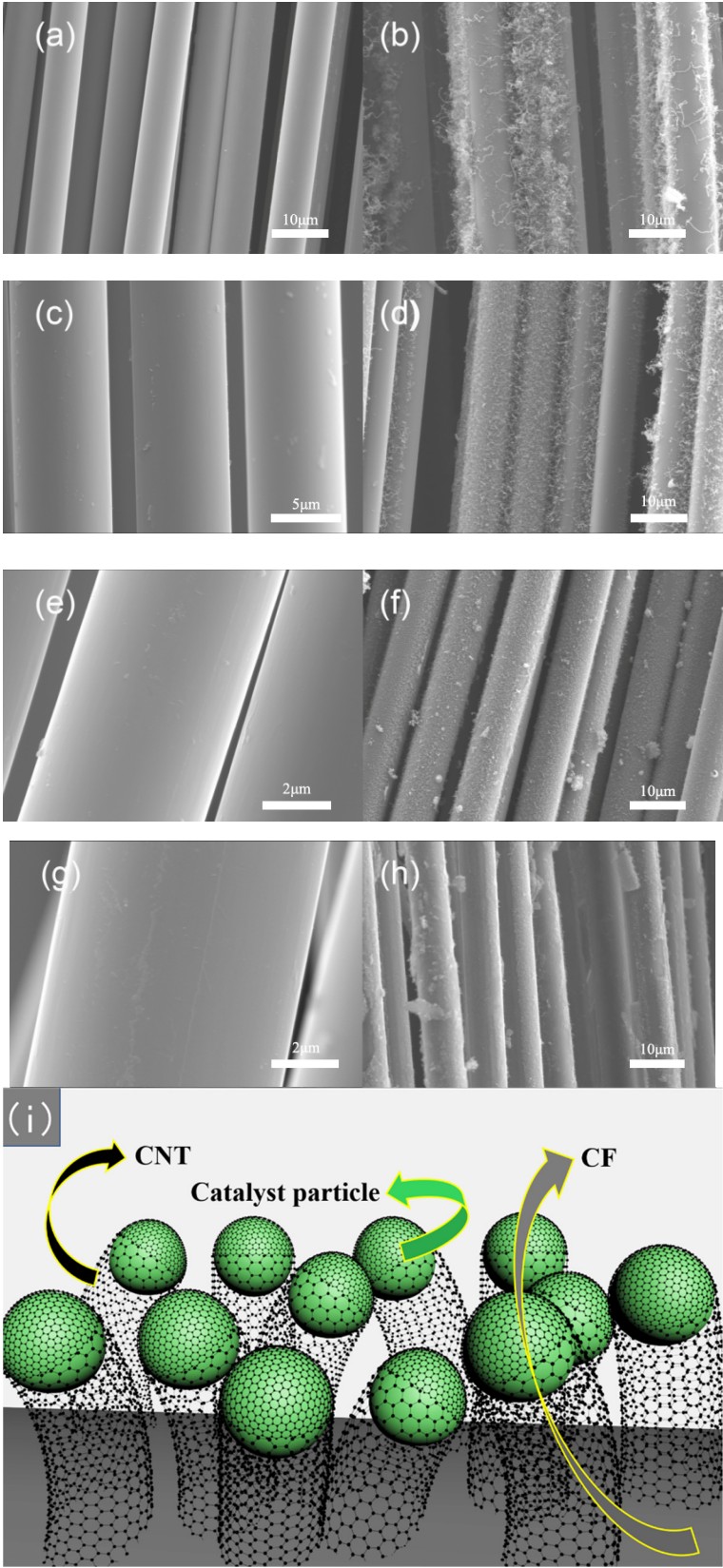

**Figure 2.** Surface morphology of (**a**,**b**) desized CFs and CFs pretreated with (**c**,**d**) HNO$_3$, (**e**,**f**) EAO, (**g**,**h**) H$_2$O$_2$ surface modification and CNTs grown on modified CFs, and (**i**) Structure diagram of CNTs/CF multi-scale reinforcements.

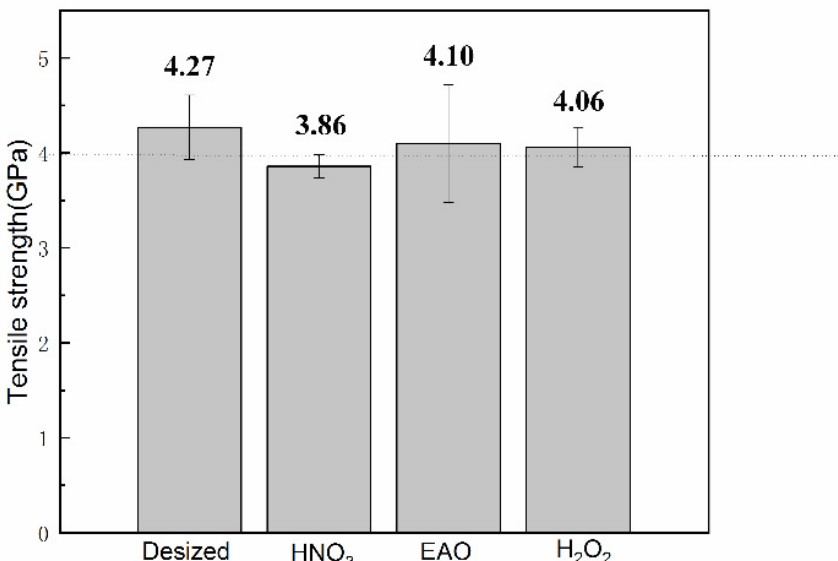

**Figure 3.** Single-filament tensile strength of CFs obtained using different surface pretreatment methods. (The dotted line in the figure indicates we set 4 GPa as the reference.)

The single-filament tensile strength of desized CFs was about 4.27 GPa. After modification, the single-filament tensile strength of each sample was reduced to varying degrees. The strength after $HNO_3$ treatment was 3.86 GPa, and that after $H_2O_2$ treatment was 4.06 GPa. Significantly, the strength after EAO treatment was 4.10 GPa, which shows that the etching on the CF surface after pretreatment with $H_2O_2$ and EAO treatment is relatively mild, while the oxidation reaction of $HNO_3$ treatment is severe, so the graphite microcrystals on the surface of CFs were significantly damaged, and the strength decreased by up to 9.6%.

By comprehensively comparing the three treatment methods, it was found that EAO treatment has great advantages. It has an obvious effect on promoting the growth of CNTs, the degree of strength reduction is acceptable, the improvement and optimization space is large, the treatment time is short, a large number of samples can be processed continuously, and the risk and economic cost are low. Therefore, EAO was selected as the pretreatment method of the CF surface and the optimization of its process conditions were explored.

### 3.3. Effect of EAO Treatment on Surface Chemical Composition of CFs

The electrochemical pretreatment method includes two processes: anodic oxidation and cathodic reduction. The input electrical energy is converted into chemical energy, and water is decomposed at the anode to generate strong oxidants to etch the CF surface, resulting in changes in chemical elements and functional groups on the surface of CFs. Qualitative and quantitative analyses were carried out by XPS tests. Table 1 shows the changes in the elemental composition of C, O, and N on the surface of CFs before and after EAO treatment.

**Table 1.** Main elemental contents of CFs before and after EAO treatment.

| Main Element | Desized (At%) | EAO (At%) |
|:---:|:---:|:---:|
| C | 92.36 | 63.52 |
| O | 7.64 | 33.83 |
| N | 0 | 2.65 |

Comparing the two samples, the elemental composition of the CF surface changed significantly after EAO treatment; notably, the oxygen content increased from 7.64% before EAO treatment to 33.83%. The desized CFs underwent high-temperature carbonization

treatment, and the nitrogen-containing functional groups on the surface were removed. After electrolysis, 2.65% nitrogen appeared on the surface of the CFs. This was due to the reaction of $NH_4H_2PO_4$ with CFs under the action of an electrical field, introducing nitrogen-containing functional groups into the CF surface.

### 3.4. Optimization of Process Parameters of EAO

EAO treatment also has two sides. On the one hand, it provides the conditions for CFs to attach catalyst precursor particles, while on the other, it causes damage and reduces the mechanical strength of CFs. In order to achieve a more balanced effect, samples with four parameters, i.e., 50 C/g, 100 C/g, 150 C/g and 200 C/g, were created for a comparative analysis. Figure 4 shows SEM images of the sample surface treated with different electrochemical intensities. It can be seen that when the intensity was 50 C/g, heavy magnification was necessary to observe the fine grooves. With the increase of intensity, the number of grooves on the CF surface gradually increased, as did the depth and width of the groove structure and the adsorption capacity for catalyst precursor; however, the excessive etching intensity penetrated deep into the body structure of CFs to cause a loss of mechanical properties.

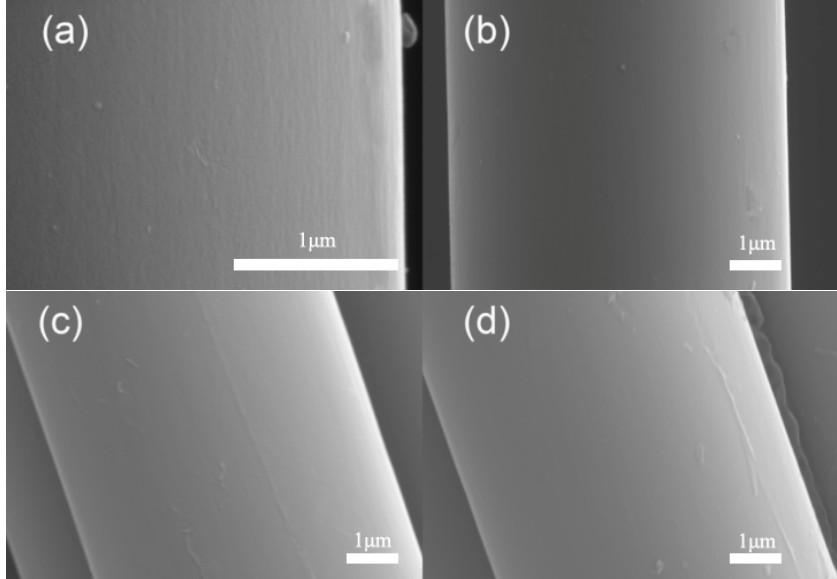

**Figure 4.** SEM images of CF surface treated with different EAO intensities. (**a**) 50 C/g, (**b**) 100 C/g, (**c**) 150 C/g, (**d**) 200 C/g.

Figure 5 presents images of CNTs grown on a CF surface treated with different EAO intensities at 550 °C. It can be seen that under the condition of 50 C/g, the number of CNTs was sparse and the distribution was uneven. Additionally, the surface of some CFs was smooth, without growing CNTs. The insufficient attracting ability of the precursor particles led to the aggregation of the precursor particles in the gaps between adjacent CFs to form larger-sized particles, so coarse carbon nanofibers emerged at the same position after the CVD process. Under the conditions of 100 C/g and 150 C/g electrochemical treatment intensity, the surface of CFs could be coated with a uniform layer of CNTs, and the surface morphology of the two was not much different. Under the condition of 200 C/g, the distribution of CNTs was too dense, and some parts demonstrated the phenomenon of clustering. This was due to the fact that the trenches obtained by high-intensity electrochemical oxidation etching were too deep, and the catalyst precursor easily aggregated, which was not conducive to the growth of CNTs. Therefore, when the EAO treatment intensities were 100 C/g and 150 C/g, the morphology of the CNTs/CF multi-scale reinforcements was optimal.

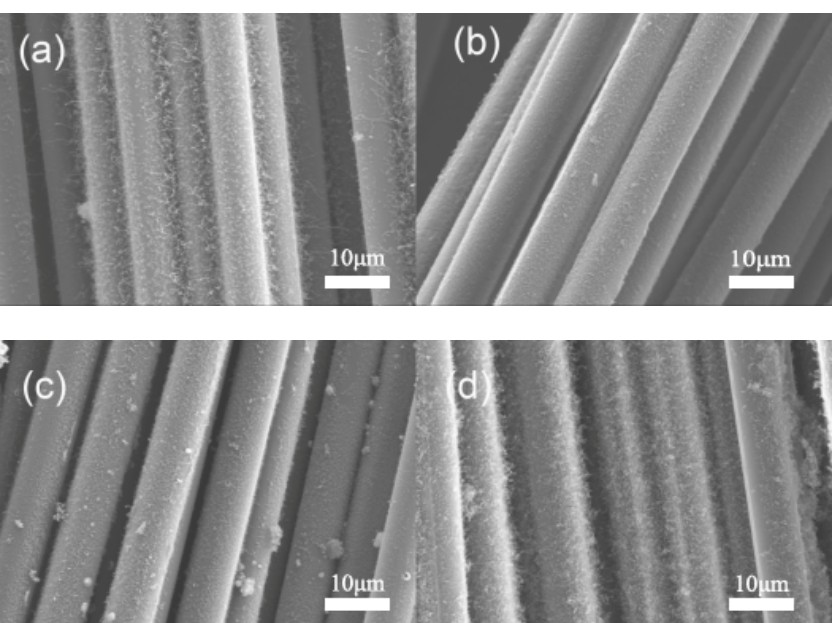

**Figure 5.** Surface morphology of CFs with CNTs treated with different EAO intensities. (**a**) 50 C/g, (**b**) 100 C/g, (**c**) 150 C/g, (**d**) 200 C/g.

Figure 6a shows the tensile strength of each sample after treatment with different electrical intensities. Compared with the value of 4.27 GPa for the desized sample, the tensile strength under the condition of 50 C/g was 4.16 GPa, i.e., a decrease of 2.57%. Additionally, corrosion occurred at this time. However, the oxidative damage to the CFs was not large. The tensile strength was 4.10 GPa at 100 C/g, a decrease of 3.98%. As the treatment intensity increased, the mechanical properties of the CFs continued to decline; the tensile strength at 150 C/g was 3.95 GPa. Under the condition of 200 C/g, the tensile strength was 3.86 GPa, marking a decrease of nearly 10%. At this time, the CFs were seriously damaged. During the electrochemical treatment process, the oxidation etching penetrated into the interior of the CFs, and the etching caused by the aggregation of catalyst particles exacerbated the strength loss.

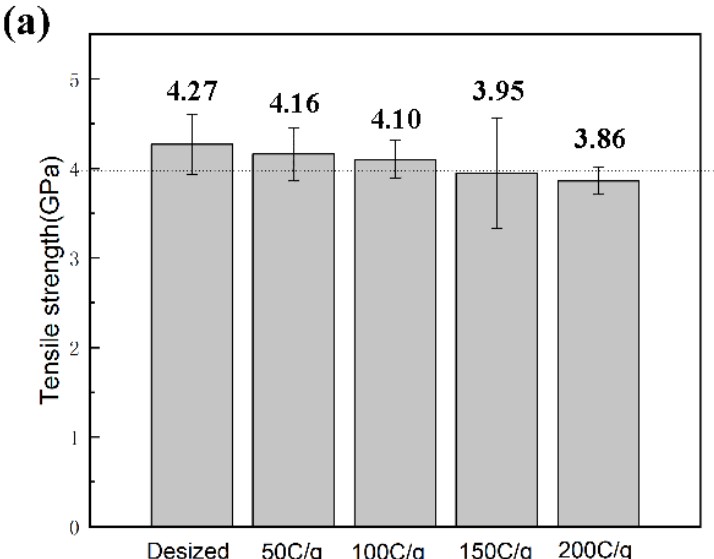

**Figure 6.** *Cont.*

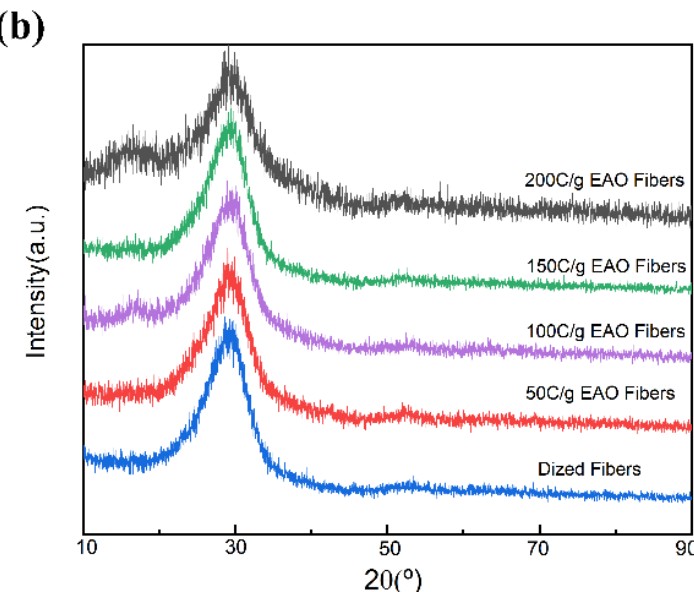

**Figure 6.** (**a**) Single-filament tensile strength (The dotted line in the figure indicates we set 4 GPa as the reference.) and (**b**) XRD curve of desized CFs and CFs treated with different EAO intensities.

It can be seen from the XPS test that the EAO treatment added functional groups containing O and N elements to the surface of CFs. These active functional groups contributed to the loading of catalyst precursor particles. These changes are reflected in the structure of the surface graphite crystallites. Figure 6b demonstrates the XRD patterns of the desized CFs and CFs with different electrochemical treatment intensities. It can be seen that each sample had a wide diffraction peak around $2\theta = 25.6°$, which is the characteristic diffraction peak of the (002) crystal plane of the CF surface through data comparison; the more complete the graphite turbostratic structure, the sharper the peak shape. Compared with the desized CFs, the peak shapes of the treated samples were broadened, indicating that the graphite structure was damaged to different degrees.

The microcrystalline structure of the (002) crystal plane of each sample was analyzed, and the parameters are illustrated in Table 2. Taking 3.354 Å as the ideal spacing of the standard graphite (002) crystal plane as the benchmark, the distance between the (002) crystal planes of each sample was closer to this value, indicating that the CF surface was less damaged and the degree of graphitization was high. The surface of CF itself had small defects, and the high temperature process of desizing damaged the surface structures of the CFs, which were different from the standard values. After electrochemical treatment, the interplanar spacing of all samples was larger than that of the desized fibers, indicating that the oxidation reaction changed the microcrystalline structure of the CF surface. Additionally, with the increase of the electrical intensity, the $d_{(002)}$ of the CF surface increased, the graphite sheet structure became increasingly loose, and the mechanical properties of the fibers decreased to a greater extent; this was also related to the tensile strength of the single-filament. The test results were consistent with these observations.

**Table 2.** Microstructure parameters of the (002) crystal plane of desized CFs and CFs treated with different EAO intensities.

| Sample | $2\theta$ (°) | $d_{(002)}$ (Å) |
|---|---|---|
| Desized | 25.693 | 3.465 |
| 50 C/g | 25.537 | 3.490 |
| 100 C/g | 25.236 | 3.504 |
| 150 C/g | 25.429 | 3.496 |
| 200 C/g | 25.512 | 3.533 |

## 4. Conclusions

CF surface pretreatment is an indispensable step before the CVD process. A $H_2O_2$ solution has a limited etching effect on the surface modification of CFs and cannot effectively increase the number of CNTs grown. A $HNO_3$ solution can effectively oxidize and etch graphite microcrystals, forming a groove structure on the surface of CFs; however, the reaction is violent, and the fiber is significantly damaged. Using the EAO processing equipment designed to oxidize and etch the surface of CFs can effectively increase the adhesion rate of the catalyst precursor, which increases the quantity of CNTs grown and significantly improves the surface morphology of CNTs/CF multi-scale reinforcements. After electrochemical treatment, the single-filament tensile strengths of 50 C/g, 100 C/g, 150 C/g, and 200 C/g samples decreased by 2.57%, 3.98%, 7.49%, and 9.60% respectively, compared with the desized CFs. The EAO method uses an oxidation reaction to etch the graphite microcrystalline structure on the surface of CFs; the higher the electrical intensity, the larger the (002) interplanar spacing of the CFs. EAO is appropriate for the pretreatment of CFs before the CVD process, which can effectively improve the loading quantity and distribution uniformity of catalyst precursor particles.

**Author Contributions:** Conceptualization, M.L., Y.W. and C.W. (Chengguo Wang); Data curation, M.L.; Investigation, M.L., Y.W., B.C. and C.W. (Chengjuan Wang); Methodology, M.L., B.C. and C.W. (Chengjuan Wang); Project administration, Y.W. and G.Z.; Software, H.T., H.J. and Z.X.; Supervision, Y.W.; Validation, M.L.; Visualization, M.L.; Writing—original draft, M.L., B.C. and C.W. (Chengjuan Wang); Writing—review and editing, M.L., Y.W., B.C., C.W. (Chengjuan Wang), H.T., H.J. and Z.X. All authors have read and agreed to the published version of the manuscript.

**Funding:** This research was funded by the Natural Science Foundation in Shandong Province (ZR2020ME134, ZR2020ME039, ZR2021ME194).

**Institutional Review Board Statement:** Not applicable.

**Informed Consent Statement:** Not applicable.

**Data Availability Statement:** All data which support the findings of this study are included within the article.

**Acknowledgments:** The authors thank the editor and the anonymous reviewers for their valuable comments on this manuscript. The authors also acknowledge the support of technical staff for assisting in preparing samples and analyzing them. This work was supported by the key research and development program of Shandong Province (2021ZLGX01).

**Conflicts of Interest:** The authors declare no conflict of interest.

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
