# Peer review of "Optimization of Electrical Intensity for Electrochemical Anodic Oxidation to Modify the Surface of Carbon Fibers and Preparation of Carbon Nanotubes/Carbon Fiber Multi-Scale Reinforcements"

_jcs, doi:10.3390/jcs6120395_

Round 1
Reviewer 1 Report
The paper address an interesting topic, i.e. the study of different technical parameters effect on the surface morphology of CNT functionalized CF. Although the description of the work is not very clear, especially in the first half of the paper, the results are interesting and no substantial deficiency can be observed. I suggest the publish the paper after the following major revisions:
· I suggest to change the paper title because it not mention the electromagnetic field intensity that is one of the main tunable parameter in the study;
· In the abstract authors use the term “loading” referred to the growth of CNT on the CF: I suggest to use the term growing to avoid misunderstanding;
· I suggest to Improve the section relative to the EAO technique because it is too short: please add more references and discuss them;
· The whole paragraph is not clearly written; I suggest to rewrite the paragraph dividing it the 3 different functionalization technique; you can use a bulleted list, a bullet for each technique;
· Why authors talk about turbostratic graphite structure? I suggest to add a reference to support that statement;
· The analysis of SEM images is not really clear; could authors explain why in figure e-f the homogeneity of the CNT is higher compared to figure a-b?
· Minor rephrasing in the text must be performed;
· Extensive English editing.
Reviewer 2 Report
This manuscript is very result-oriented. Authors used different method to etch the CFs’ surface, and found that EAO method give rise to relatively obvious groove structure, leading to evenly distribution of Ni on the surface of CFs. From the XPS results, CFs’ surface has been largely destroyed and exhibits defects. It also explains why tensile property has been slightly reduced. A few questions are listed below:
Can electrochemical anodic oxidation (EAO) be easily scalable?
What is the chemical composition of sizing agent?
What is the Ni catalyst particle size? Are you able to characterize the CNTs grew on the surface?
Authors need to load resultant CFs into composite material and them investigate whether or not properties (mechanical, thermal…) can be improved.
Round 2
Reviewer 1 Report
After the asked corrections the paper can be published.
Reviewer 2 Report
Authors gave sufficient answers. If the composite material could be made and corresponding results could be published, it would be great.